# Treatment-Resistant Depression in Poland—Epidemiology and Treatment

**DOI:** 10.3390/jcm11030480

**Published:** 2022-01-18

**Authors:** Piotr Gałecki, Jerzy Samochowiec, Magdalena Mikułowska, Agata Szulc

**Affiliations:** 1Department of Adult Psychiatry, Medical University of Lodz, 91-229 Lodz, Poland; 2Department of Psychiatry, Pomeranian Medical University, 71-460 Szczecin, Poland; jerzy.samochowiec@pum.edu.pl; 3Janssen Pharmaceutical, 02-135 Warsaw, Poland; MMikulow@ITS.JNJ.com; 4Department Psychiatry, Faculty of Health Sciences, Medical University of Warsaw, Partyzantów 2/4, 05-802 Pruszków, Poland; agata.szulc@wum.edu.pl

**Keywords:** major depressive disorder (MDD), treatment-resistant depression (TRD), epidemiology

## Abstract

(1) Background: Major depressive disorder (MDD) is one of the most prevalent psychiatric disorders worldwide. Although several antidepressant drugs have been developed, up to 30% of patients fail to achieve remission, and acute remission rates decrease with the number of treatment steps required. The aim of the current project was to estimate and describe the population of treatment-resistant depression (TRD) patients in outpatient clinics in Poland. (2) Methods: The project involved a representative sample of psychiatrists working in outpatient clinics, chosen through a process of quota random sampling. The doctors completed two questionnaires on a consecutive series of patients with MDD, which captured the patients’ demographics, comorbidities, and medical histories. TRD was defined as no improvement seen after a minimum of two different antidepressant drug therapies applied in sufficient doses for a minimum of 4 weeks each. The data were weighted and extrapolated to the population of TRD outpatients in Poland. (3) Results: A total of 76 psychiatrists described 1781 MDD patients, out of which 396 fulfilled the criteria of TRD. The TRD patients constituted 25.2% of all MDD patients, which led to the number of TRD outpatients in Poland being estimated at 34,800. The demographics, comorbidities, medical histories, and histories of treatment of Polish TRD patients were described. In our sample of the TRD population (mean age: 45.6 ± 13.1 years; female: 64%), the patients had experienced 2.1 ± 1.6 depressive episodes (including the current one), and the mean duration of the current episode was 4.8 ± 4.4 months. In terms of treatment strategies, most patients (around 70%) received monotherapy during the first three therapies, while combination antidepressant drugs (ADs) were applied more often from the fourth line of treatment. The use of additional medications and augmentation was reported in only up to one third of the TRD patients. During all of the treatment steps, patients most often received a selective serotonin reuptake inhibitor (SSRI) and a serotonin norepinephrine reuptake inhibitor (SNRI). (4) Conclusions: TRD is a serious problem, affecting approximately one fourth of all depressive patients and nearly 35,000 Poles.

## 1. Introduction

Major depressive disorder (MDD) is one of the most prevalent psychiatric disorders worldwide, especially in high-income regions [1,2,3,4,5]. It causes individual suffering, a loss of productivity and increased health care costs; it is also associated with a high suicide risk [2]. As such, MDD represents a major economic and medical burden for societies [1]. Depressive disorders directly account for 4.4% of the disease burden worldwide and 7.2% in the European Union [6]. According to WHO estimates, in 2015, depressive disorders occurred in 322 million people worldwide and 40.3 million in Europe [7]. In Poland, around 1.5 million people suffer from depression, and it is estimated that approximately 3% of people of productive age (i.e., 766,000 adult Poles) had at least one depressive episode in their lives [8,9].

Over the last 60 years, several antidepressant drugs (ADs) have been developed that can be effective in treating MDD. Although treatment leads to significant improvement in several patients, up to 30% of them fail to achieve remission [1,2,4,10]. Even after sequential treatments, 10–20% of depressive patients remain significantly symptomatic [4]. The results of the Sequenced Treatment Alternatives to Relieve Depression (STAR*D) trial of 3671 depressive outpatients suggested remission rates for patients after the first through fourth treatments of 37%, 31%, 14%, and 13%, respectively [11]. 

Treatment-resistant depression (TRD) is typically limited to unipolar depression [12]. Many definitions of TRD have been proposed during the past few decades, including the failure of a single AD treatment, the failure of three or more treatments (including tricyclic antidepressants (TCAs)) or of five or more treatments, a failure to respond to electroconvulsive therapy (ECT), the failure of a single TCA treatment, or of a single therapy regimen using heterocyclic antidepressants [13,14,15]. Some TRD models aimed to classify levels of treatment resistance; for example, Thase and Rush’s model proposed five levels of resistance, from the failure of more common therapies to a lack of response to ECT [16]. Currently, TRD is most often defined as an inadequate response to at least two treatments with adequate dosages and durations [4,10,17]. Souery, for example, proposed operational criteria for TRD defined as a failure to respond to two adequate trials of different classes of AD, that is, consecutive treatments with two different ADs, each administered separately at an adequate dosage for a period of 6 to 8 weeks [18]. Different stages of treatment resistance correspond to the number of failed AD treatments [18]. Additionally, the European Medicines Agency defined treatment resistance as a nonresponse to at least two adequate ADs of the same or different classes at adequate dosages for adequate durations and with an adequate affirmation of treatment adherence [1]. 

Several risk factors for TRD have been described, including comorbid medical or psychiatric conditions (e.g., hypothyroidism, vitamin deficiencies, anxiety, or personality disorders), clinical factors (e.g., illness severity or delay in initiating treatment), drug interactions, genetic factors, sociodemographic factors (e.g., gender or education level), and external factors supporting illnesses [5,10,12,19]. It has also been identified that bipolarity is a significant factor of treatment resistance due to the inadequate treatment of bipolar diathesis in depression diagnosed as unipolar [10,20,21].

TRD is associated with a higher risk of comorbidity for both physical and mental disorders, the impairment of functioning—including social functioning—lower quality of life, and increased risk of suicide [1,2,3,17]. It is highly recurrent, with approximately 80% of TRD patients relapsing within a year of remission [3]. 

TRD patients generate higher medical costs (both direct and indirect) than MDD patients who are responsive to treatment; these costs increase with TRD severity [2,12,22,23]. TRD patients have at least 12% more outpatient visits, use 1.4 to 3 times more psychotropic medications and have twice as high a risk of hospitalization as other depressive patients [22]. The total medical costs for hospitalized TRD patients are 6 times higher than those of non-treatment-resistant depressive patients [22]. The costs of a TRD episode have been estimated to be 2.7–5.8 times higher than corresponding costs for a non-TRD episode because of longer episode duration, greater frequency of usage, and higher costs per visit [23]. Total depression-related costs for TRD patients have been estimated to be 19 times higher than for other depressive patients [22]. In the USA, up to 47% of depression-associated costs are attributable to treatment resistance [2,24]. In Poland, the direct costs of depression exceed 1 billion PLN, while the indirect costs have been estimated at 1–2.6 billion PLN [25,26]. However, TRD treatment’s share in these costs is unknown because there is a lack of information about the size and characteristics of the Polish population of TRD patients. 

TRD has never been specifically addressed by researchers in Poland. Thus, the treatment habits of Polish psychiatrists in depressive patients after the failure of two lines of treatment are unknown. The aim of the current project was to estimate and describe the population of TRD patients in outpatient clinics in Poland. The project bridges important knowledge gaps, providing doctors, payers, and other stakeholders with a full perspective and overview of Polish depressive patients’ histories of prior treatments.

## 2. Materials and Methods

The data were collected between October 2020 and May 2021 as part of the Economedica project, “Treatment-resistant depression”, including psychiatrists working in outpatient clinics in Poland and treating depression. Two-stage sampling was used in the project. 

### 2.1. Selection of Psychiatrists

The sampling of the doctors reflected the regional distribution of psychiatric outpatient clinics in Poland and considered their size of contract for psychiatric care and addiction treatment. First, clinics were randomly chosen from all outpatient clinics offering treatment financed by the National Health Fund. If the doctors refused to take part in the study, another clinic was chosen according to its geographical proximity to the original one. Psychiatrists who were only employed in private clinics (constituting 10.3% of all psychiatrists working in ambulatory health care according to the Health Data Management basis) were not included. However, they were considered during the process of data weighting and extrapolation. In spite of the fact that the invited doctors were asked to describe patients from both public and their own practice (if practicing in both), there might be potential selection bias, as psychiatrists working independently in their own practice alone may see less severely ill patients.

### 2.2. Selection of Patients

The psychiatrists who agreed to take part in the study filled in two questionnaires. Questionnaire A included the following:A record of all adult patients they examined within the last 4 weeks, in both public and private clinics;A short description of depressive patients examined within the last 8 weeks (maximum last 30), according to the chronological order of the examinations.The inclusion criteria for depressive patients were as follows:An adult patient (at least 18 years old);Currently in an active episode of unipolar depression or in their first depressive episode, regardless of the treatment stage or effectiveness.Questionnaire B was used to collect more detailed information regarding the patients in an active episode of TRD. The inclusion criteria were as follows:An adult patient (at least 18 years old);Currently in a depressive episode (including either the first or a subsequent episode of recurrent depressive disorder, F32 and F33);The current episode fulfils the criteria of TRD used in this study: no improvement seen after at least two different AD therapies of sufficient dosage for at least 4 weeks each.

Sufficient dosages were defined by the experts of the Scientific Council of the project. Due to the definition of TRD applied in the study, the minimum treatment duration for a depressive episode was 56 days.

The patients who had been examined in the 2 months preceding data collection were described retrospectively, based on their medical documentation. Each psychiatrist provided data for 1–13 patients, chosen chronologically. A detailed description of the project is presented in Table 1. 

### 2.3. Data Extrapolation

To formulate general conclusions regarding the Polish population of patients in an active episode of unipolar TRD treated in outpatient clinics, the data collected were weighted and extrapolated. The following data collected in Questionnaire A of the project and the National Health Fund registers [27] were used:The overall number of patients examined because of depression in mental health clinics financed with public funds (“NHF on health—depression, February 2020”);The number of visits to mental health clinics financed with public funds because of depression (“NHF on health—depression, February 2020”);The proportion of medical visits in private clinics (“Economedica—patients record”; Questionnaire A);The mean number of psychiatric visits per 12 months (“Economedica—patients record”; Questionnaire A);The characteristics of adult depressive outpatients (“Economedica—patients record”; Questionnaire A).

The analysis allowed for (1) an estimation of the size of the Polish population of adult outpatients in an active episode of TRD and (2) the definition of weights for extrapolating the data collected in Questionnaire B to the Polish population of outpatients in an active episode of TRD.

The statistical analysis was performed by PEX PharmaSequence using SPSS 20.0. 

## 3. Results

Overall, 76 psychiatrists took part in the project. Out of the 1781 patients described in Part A of the questionnaire as having an active depressive episode, 396 fulfilled the definition of unipolar TRD used in this study and were included in the analysis. Their mean ± standard deviation age was 45.6 ± 13.1 years, median 44.0; 64% of them were women; 62% were employed (50% full-time and 12% part-time). Over half (58%) of the patients had co-existing diseases, mainly high blood pressure (32%), obesity (16%), hypothyroidism (14%), lipid metabolism disorders (12%), and pain (11%). A portion were addicted to alcohol (5%), sedatives, sleeping pills, or pain medication (6%) or narcotics (2%). About half (51%) had an anxiety disorder, while 13% had a personality disorder, 1% a psychotic disorder and 4% other psychiatric disorders. In 75% of cases, the patients had sought treatment from the psychiatrists on their own or on their relatives’ initiative.

### 3.1. Medical History

Detailed characteristics of the research sample are presented in Table 2. On average, the patients had experienced 2.1 ± 1.6 depressive episodes, median 2.0, including the current one. About one third (35.6%) of them were experiencing their first episode, while for another 40.1%, the current episode was at least their second; for the remaining 24.2% it was difficult to say which episode they were experiencing. The mean duration of the current episode was 4.8 ± 4.4 months, median 3.1. In 27.0% of the patients, the current episode had lasted between 2 and 3 months; in another 48.5% it had lasted between 3 and 6 months; in the remaining 24.5% the episode was longer than 6 months. The first episode occurred at a mean age of 40.7 ± 12.1 years, median 40.0. Most patients (52.0%) had experienced their first episode before the age of 40. Out of the 255 of patients who had experienced more than one episode, the episodes occurred less than once a year in 46.3% of them. One episode per year was reported for 32.5% of the patients, while 14.1% of them had two or three episodes per year. Out of the entire sample, 13.1% were hospitalized during the previous 2 years and 7% during the current episode. The most common reasons for hospitalization were to optimize pharmacotherapy (59.6%), because of a suicide attempt or suicidal thoughts (36.5%), and for psychotherapy (30.8%). The last hospitalization due to a depressive episode lasted on average 8.6 ± 7.0 weeks, median 8.00, with most patients (67.3%) being hospitalized for 1 to 3 months. One third (33.1%) of the patients reported having suicidal thoughts during the last episode, while 7.1% had tried to commit suicide (1% during the current episode).

### 3.2. Treatment

Approximately half (51.8%) of the patients included in the study were in the midst of their second medical treatment, while 43.7% were having therapy for the third time. 

During the first three therapies, most patients (around 70%) received monotherapy. Combination AD therapy was applied more often in the fourth (44.5%) and fifth (66.7%) treatments. The use of additional medications and augmentation increased with each subsequent therapy but was reported in only up to one third of the patients (Table 3). Other types of therapies, including ECT, transcranial magnetic stimulation, vagus nerve stimulation, etc. were inaccessible.

In most cases, the patients received a selective serotonin reuptake inhibitor (SSRI; 83.6%) and a serotonin norepinephrine reuptake inhibitor (SNRI; 72.2%, Table 4).

During the first-line antidepressant therapy in the current major depressive episode (MDE), the most-often applied medications were sertraline (17%) and escitalopram (15.9%, Table 5). 

During the second-line treatment, venlafaxine (18.7%) and duloxetine (13.6%) were the most-often prescribed ADs (Table 6). 

During the third-line therapy as well, venlafaxine (19.4%) and duloxetine (17.8%) were used most often (Table 7).

During the fourth-line antidepressant therapy in the current MDE, which applies to 18 patients, the most often prescribed medications included venlafaxine (four patients) and duloxetine (two patients). Other therapies (trazodone with agomelatine, duloxetine or escitalopram; clomipramine; vortioxetine; bupropion; sertraline with agomelatine; venlafaxine/venlafaxine XR with mirtazapine, trazodone or vortioxetine and bupropion; escitalopram alone or with mirtazapine) were prescribed to one patient each. In the fifth therapy, each of the six patients received a different AD, including venlafaxine, vortioxetine, and vortioxetine with trazodone, as well as combinations of escitalopram and bupropion, sertraline, bupropion, and mirtazapine or opipramol and trazodone. 

In addition to pharmacotherapy, the patients underwent different forms of psychotherapy and psychoeducation (Table 2).

### 3.3. Estimation of the Total Number of TRD Patients in Outpatient Clinics in Poland

The proportions of different subgroups of depressive patients in the sample are shown in Figure 1. We extrapolated these values to the total population of Polish depressive outpatients by first estimating the population size of all depressive outpatients in Poland using the following data:

The prevalence of depression in Poland, estimated at 2.8% in 2017, according to Global Burden of Disease, conducted by the Institute for Health Metrics and Evaluation (IHME, 2020);

The number of patients diagnosed with affective disorders (F31.3–F31.6, F32, F33, F34.1, F34.8, F34.9, F38, and F39 according to the ICD-10) who visited psychiatric clinics financed with public funds at least once in 2020: 259,038, according to National Health Fund registers.

Based on the above values, the number of patients being treated outside of the public health care system was estimated to be 41.1%. This estimation, based on extensive population data, was made because the exact proportion of depressive patients being treated outside of the public health care system is unknown, though it constitutes an important number for depressive disorders. 

On this basis, the total number of depressive outpatients in Poland was estimated at 440,000. Using this value and the proportions obtained in the current study, we estimated the total number of unipolar TRD patients in outpatient clinics in Poland to be 34,800. TRD patients constituted 25.2% of all depressive patients in an active depressive episode.

The collected data were weighted to obtain proportions that describe the Polish population of TRD outpatients. We estimated that 66% of the patients are women, the mean age of patients is 45.3 ± 12.4 years, median 44.0, and 60% of them have co-existing diseases. The first depressive episode occurs at the mean age of 40.5 ± 11.5 years, median 40.0. One in ten (13%) of them have been hospitalized in the last 24 months (7% during the current episode), and their hospitalization lasted on average 8.4 ± 6.7 weeks, median 8.0. One third (36%) of them had suicidal thoughts and 6% of them attempted suicide. Detailed results are presented in Table 2. 

## 4. Discussion

### 4.1. TRD Prevalence

To our knowledge, our study is the first to quantify and describe the population of TRD outpatients in Poland. We found that the proportion of TRD patients among the adult depressive outpatients in an active episode was 25.2%. A direct comparison of this result to other findings is difficult due to the inconsistencies in defining TRD across various studies. Moreover, it has been suggested that the proportion of TRD patients is usually higher in studies based on clinical data sets than in studies that used administrative data [28]. Nevertheless, our finding is consistent with other studies. The prevalence of TRD—defined as no or minimal improvement following two or more AD trials lasting a minimum of 6 weeks—in Canada has been estimated at 21.7% [29]. In Latin America, a prevalence of 29% for TRD was reported among all MDD patients and 32% among treated MDD patients, with the exact values for countries ranging from 21% in Mexico to 40% in Brazil [30]. An analysis of National Health and Wellness Survey data from five European countries found a prevalence of 18.8% for TRD [31]. In Taiwan, 21% of patients treated for MDD were then classified as having TRD [32]. In the USA, 30.9% of treated MDD patients were classified as having TRD [24]. The much lower (8.3% in Hungary and 6.6% in the USA) as well as the much higher (55% in the UK) TRD prevalence reported in the literature can be explained by methodological differences in the inclusion criteria, the definition of TRD, or the source of the data [23,28,33]. Our results, being in line with previous reports, suggest that TRD is a serious problem that affects approximately one fourth of all depressive patients and nearly 35,000 Poles.

### 4.2. TRD Treatment

Most of the TRD patients in our sample were prescribed AD monotherapy; the proportion of AD combinations increased after the third failed therapy. This result confirms the findings of other authors, showing that most TRD patients are prescribed a single AD [23,33]. In one study, 57% of the TRD patients took two ADs as opposed to 43% following an AD monotherapy; however, no information regarding the number of previous therapies was provided, so it is possible that some of the patients were under subsequent treatments, where the probability of being administered combination therapies increased [29]. 

Although the combination of two ADs is often used in TRD treatment, evidence regarding its efficacy is sparse [1,6]. There are some findings that show better treatment outcomes for combinations of bupropion and SSRI; reboxetine and an SSRI; mirtazapine or mianserin and an SSRI or an SNRI; monoamine oxidase inhibitor (MAOI) and TCA; or fluoxetine and desipramine, as compared to monotherapy [34,35,36,37,38].

In our study, TRD patients were most frequently prescribed SSRI and SNRI therapies. This is in line with other studies mentioning SSRIs and SNRIs as the first-line treatment for MDD [4,6] as well as with the guidelines of the Polish Psychiatric Association and the National Consultant for Adult Psychiatry for pharmacological treatment of a depressive episode and recurrent depressive disorder [39]. Other studies mainly point to SSRIs as the predominant therapy in TRD, with SNRIs being prescribed less often [23,30,32,33]. SSRIs are considered to be the safest AD [39]. 

In our study, sertraline and escitalopram were the most-often prescribed medications in the first treatment, while venlafaxine and duloxetine comprised the most common combination in the second and third. This finding is in line with Kubitz and Rizvi’s results, showing that sertraline, venlafaxine XR, citalopram, bupropion XL, fluoxetine, and escitalopram were the most-often prescribed ADs and that venlafaxine XR, escitalopram, bupropion XL, and duloxetine were used most often when switching medications [23,29]. It is also in line with the recommendations of French experts that an SSRI should be replaced with duloxetine and venlafaxine in the case of a failed initial treatment [6]. Switching to a class of AD with a different mechanism of action is often recommended, although there is no compelling evidence that this approach is more effective than within-class switching [12]. Some studies have demonstrated superior treatment effects for between-class switches [4,40], while others did not observe such an effect [41].

Regarding ADs, there is evidence that venlafaxine is more effective than SSRIs [3], while a recent meta-analysis reported that among the 12 ADs compared, sertraline, escitalopram, venlafaxine, and mirtazapine had the highest efficacy and sertraline and escitalopram were the best tolerated [42]. The STAR*D study, which compared the effectiveness of switching to bupropion, sertraline, and venlafaxine in patients who did not achieve remission after citalopram treatment, found no significant differences between these three ADs [43]. 

In our study, only between 24% and 33% of TRD patients were prescribed any additional therapy, with the likelihood of augmentation increasing with subsequent therapies. This finding is in line with the observation from a UK study that augmented AD treatment was rare in TRD patients [33]. However, evidence exists that augmenting an AD with certain agents is effective in TRD treatment. Good effects were observed for quetiapine [44,45,46], aripiprazole [15,47,48,49], olanzapine [50], and risperidone [51,52] in combination therapy with SSRIs/SNRIs. A meta-analysis comparing 48 randomized, controlled trials using 11 different augmentation agents demonstrated the effectiveness of aripiprazole, lithium, quetiapine, and T3 [53]. Another meta-analysis demonstrated the positive effect of aripiprazole, olanzapine, quetiapine, and risperidone over a placebo [54]. Olanzapine–fluoxetine combination therapy was shown to be more effective than both olanzapine and fluoxetine monotherapy in one study [55], but not better than fluoxetine or venlafaxine monotherapy in another [56]. There is also good evidence for the efficacy of lithium as an augmentation agent in TRD [1,10].

### 4.3. Strengths and Limitations

Our study has some limitations. First, we did not compare the characteristics of TRD patients to non-treatment-resistant ones. However, the main aim of the study was to estimate and describe the population of TRD patients and their treatment. 

TRD was defined in our project as the first or subsequent episode of MDD that does not respond to treatment despite a minimum of two different AD therapies of sufficient dosage for a minimum of 4 weeks each. However, no objective measure of symptom severity was used, while the preferred outcome measure in studies on depression is the remission of symptoms using a standardized and validated test [57]. There are several scales that can be used to evaluate symptom severity in depression, for example, the Hamilton depression rating scale (HAM-D), the Montgomery–Asberg depression rating scale (MADRS), and clinical global impressions (GRADE-high) [58,59,60]. Response to treatment is usually defined as a 50% reduction in symptoms, while improvement means a 20–30% reduction in scores [5,57,61]. Despite the existence of validated measures of depressive symptom severity, Rizvi reported that only 25% of doctors in their study used them. While some studies on TRD use standardized scales to assess improvement [28,31,62], others use doctors’ evaluations [29] or data regarding treatment found in administrative databases [23,31,32,63]. Our project was based on real-world evidence, and its aim was to reflect decisions made by psychiatrists in practice. Therefore, treatment failure was defined by the physicians themselves, based on either standardized scales or subjective evaluations of them. Some studies have suggested that treatment efficacy should be evaluated after 6 to 8 or even 10 to 12 weeks [19]. We used 4 weeks as a minimum treatment duration, which is consistent with other studies [29,58]; it has been shown that a lack of improvement after 4 weeks predicts poorer outcome at 8 weeks [19,29]. It is possible, however, that using longer-duration treatments would result in slightly different estimations. 

The strength of our study was the structured questionnaires used to collect patient data, while several studies only used administrative databases or physicians’ reports to define TRD patients. We also defined the minimal doses of ADs, creating precise criteria for adequate treatment.

## Figures and Tables

**Figure 1 jcm-11-00480-f001:**
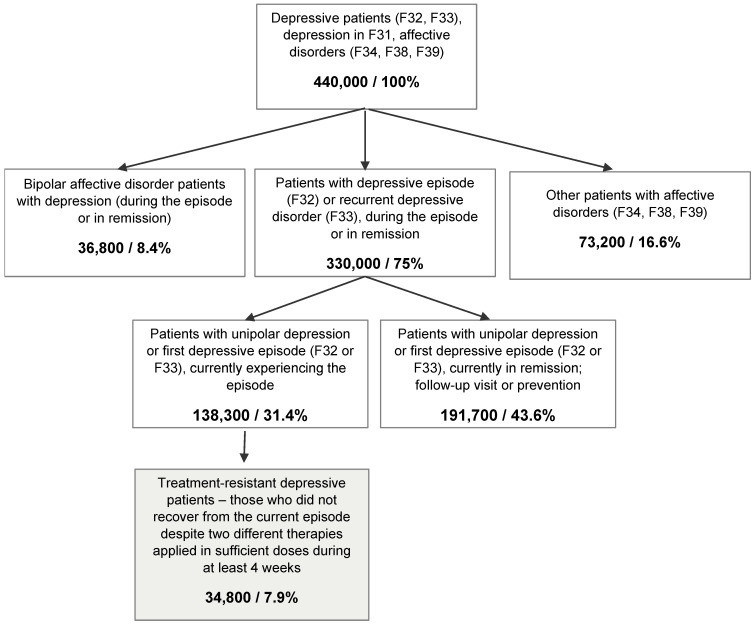
Estimated population sizes of different groups of depressive patients in Poland; percentages represent proportions of the total sample.

**Table 1 jcm-11-00480-t001:** Data collection methods for Questionnaires A and B.

	Questionnaire A Visits’ Records	Questionnaire B Detailed Descriptions of Patients
Data collection type	Retrospective	
Data collector	Psychiatrists working in an outpatient clinic	
Data source	Medical documentation	
Population	Adult patients in active episodes of unipolar depression or in their first depressive episode	Adult patients in an active episode of treatment-resistant depression
Aim	To collect data about the number of patients with forms of unipolar depression	To collect detailed data on demographics, course of disease, and treatment;to extrapolate to the Polish population
Topics/questions	Record of adult patients examined in the last 4 weeks in all outpatients’ clinics where they worked as physiatrists Typical number of depressive patients examined this month every year A short description of patients in an active depressive episode examined during the last 8 weeks (maximum last 30 patients)	Patients’ profile Patients’ current state Current treatment Treatment history of the current episode—a maximum of 24 months

**Table 2 jcm-11-00480-t002:** Medical characteristics of TRD patients included in the study.

Characteristic	Number of Patients	Proportion of Patients (Unweighted Data)	Polish Population Characteristics (Weighted Data)
Reason for seeing the psychiatrist			
Patients’ or their relatives’ initiative	297	75.0%	74.8%
GP	68	17.2%	16.2%
Other specialist	24	6.1%	6.2%
Depression diagnosed during the treatment of another disease	7	1.8%	2.8%
Number of episodes (excluding “hard to say” answers)			
1 episode	141	35.6%	36.4%
2 episodes	62	15.7%	15.2%
3 episodes	54	13.6%	15.3%
4–13 episodes	43	10.9%	11.2%
Age of the first episode (years, excluding no data)			
≤25	37	10.5%	9.4%
26–30	37	10.5%	11.2%
31–35	40	11.3%	12.9%
36–40	70	19.8%	20.5%
41–45	55	15.5%	15.3%
46–50	50	14.1%	13.4%
51–55	27	7.6%	7.3%
56–60	17	4.8%	4.9%
61–65	7	2.0%	1.8%
66–70	7	2.0%	1.8%
>70	7	2.0%	1.5%
Number of episodes per year			
Fewer than 1	118	46.3%	46.8%
1	83	32.5%	31.3%
2	33	12.9%	15.1%
3	3	1.2%	1.5%
Other	18	7.1%	5.3%
The reason for hospitalization			
Optimization of pharmacotherapy	31	59.6%	66.4%
Suicide attempt or suicidal thoughts	19	36.5%	37.3%
Psychotherapy	16	30.8%	28.8%
Risk of deterioration of mental health	12	23.1%	21.1%
Patients’ inability to satisfy their basic needs by themselves	4	7.7%	10.1%
Electroconvulsive therapy	2	3.8%	3.8%
Other depression-related reason	2	3.8%	2.7%
Duration of hospitalization			
1 week	2	3.8%	4.4%
2–3 weeks	7	13.5%	11.0%
1 month	10	19.2%	21.9%
>1–2 months	13	25.0%	25.8%
>2–3 months	12	23.1%	21.7%
>3 months	8	15.4%	15.3%
Duration of the current episode			
2–3 months	107	27.0%	29.3%
>3–6 months	192	48.5%	49.5%
>6–12 months	67	16.9%	14.3%
>12–18 months	23	5.8%	4.6%
>18–24 months	5	1.3%	1.9%
>24 months	2	0.5%	0.5%
Number of therapies during the current episode within last 24 months			
2	205	51.8%	54.9%
3	173	43.7%	39.9%
4	12	3.0%	4.0%
5	6	1.5%	1.2%
Additional forms of therapy			
Individual psychotherapy or cognitive behavioral therapy	154	38.9%	40.5%
Group psychotherapy	19	4.8%	4.4%
Family therapy	3	0.8%	0.6%
Psychoeducation	90	22.7%	23.2%
Art therapy, music therapy, or occupational therapy	6	1.5%	1.7%
Other forms of therapy	8	2.0%	1.8%
None	176	44.4%	43.9%

**Table 3 jcm-11-00480-t003:** Medications used in each therapy.

	Therapy									
	1st		2nd		3rd		4th		5th	
1 antidepressant	278	70.4%	276	69.7%	133	69.6%	10	55.6%	2	33.3%
2 antidepressants	116	29.4%	114	28.8%	54	28.3%	7	38.9%	3	50.0%
3 antidepressants	1	0.3%	6	1.5%	4	2.1%	1	5.6%	1	16.7%
Additional therapy	95	24.0%	106	26.8%	59	30.9%	6	33.3%	2	33.3%

**Table 4 jcm-11-00480-t004:** Frequency of usage of different antidepressant types.

Antidepressant Medication Type	Number of Patients	Frequency of Usage
SSRI	331	83.6%
SNRI	286	72.2%
Tricyclic antidepressants	42	10.6%
MAOI	4	1.0%
Other antidepressants	248	62.6%

MAOI—monoamine oxidase inhibitor; SNRI—serotonin–norepinephrine reuptake inhibitors; SSRI—selective serotonin reuptake inhibitor.

**Table 5 jcm-11-00480-t005:** Administered first-line antidepressant treatment during the current MDE (therapies with the same frequency of application are grouped together).

Antidepressants Used during the First Therapy (*n* = 396).	Number of Patients	Usage Frequency
Sertraline	67	17.0%
Escitalopram	63	15.9%
Venlafaxine/venlafaxine XR	30	7.6%
Sertraline + trazodone	24	6.1%
Paroxetine	21	5.3%
Fluoxetine	20	5.1%
Escitalopram + trazodone	18	4.6%
Citalopram Mianserin	15	3.8%
Tianeptine	11	2.8%
Duloxetine	9	2.3%
Escitalopram + mirtazapine Trazodone	6	1.5%
Citalopram + trazodone Escitalopram + mianserin Venlafaxine/venlafaxine XR + trazodone	5	1.3%
Clomipramine Opipramol Sertraline + mianserin Venlafaxine/venlafaxine XR + mianserin	4	1.0%
Agomelatine Citalopram + mianserin Duloxetine + mirtazapine Fluoxetine + mianserin Fluoxetine + trazodone Venlafaxine/venlafaxine XR + mirtazapine	3	0.8%

**Table 6 jcm-11-00480-t006:** Administered second-line antidepressant treatment during the current MDE (therapies with the same frequency of application are grouped together).

Antidepressants Used during the Second Therapy (*n* = 396)	Number of Patients	Usage Frequency
Venlafaxine/venlafaxine XR	74	18.7%
Duloxetine	54	13.6%
Sertraline	24	6.1%
Escitalopram	22	5.6%
Fluoxetine	18	4.5%
Venlafaxine/venlafaxine XR + trazodone	15	3.8%
Paroxetine Trazodone	14	3.5%
Duloxetine + trazodone	13	3.3%
Mirtazapine Vortioxetine	11	2.8%
Clomipramine	9	2.3%
Citalopram Duloxetine + mianserin Duloxetine + mirtazapine Venlafaxine/venlafaxine XR + mianserin	8	2.0%
Mianserin Sertraline + trazodone	7	1.8%
Venlafaxine/venlafaxine XR + mirtazapine	6	1.5%
Escitalopram + trazodone	5	1.3%
Paroxetine + trazodone	4	1.0%
Agomelatine	3	0.8%
Citalopram + mirtazapine Escitalopram + mianserin Fluoxetine + trazodone Mianserin + venlafaxine/venlafaxine XR Moclobemide Sertraline + mianserin Sertraline + mirtazapine	2	0.5%

**Table 7 jcm-11-00480-t007:** Administered third-line antidepressant treatment during the current MDE (therapies with the same frequency of application are grouped together).

Antidepressants Used during the Third Therapy (*n* = 191)	Number of Patients	Usage Frequency
Venlafaxine/venlafaxine XR	37	19.4%
Duloxetine	34	17.8%
Bupropion	9	4.7%
Mirtazapine	8	4.2%
Paroxetine	7	3.7%
Clomipramine Sertraline Vortioxetine	6	3.1%
Venlafaxine/venlafaxine XR + mirtazapine Venlafaxine/venlafaxine XR + trazodone	5	2.6%
Citalopram Duloxetine + mirtazapine Duloxetine + trazodone Escitalopram Fluoxetine Vortioxetine + trazodone	4	2.1%
Duloxetine + mianserin	3	1.6%
Bupropion + trazodone Escitalopram + trazodone Mianserin Moclobemide Venlafaxine/venlafaxine XR + mianserin	2	1.0%
Agomelatine Citalopram + venlafaxine/venlafaxine XR Duloxetine + bupropion Duloxetine + trazodone + opipramol Escitalopram + mirtazapine Fluoxetine + clomipramine Fluoxetine + mirtazapine Fluoxetine + trazodone + opipramol	1	0.5%

## Data Availability

The data that support the findings of this study are available from the corresponding author upon reasonable request.

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
