# Peer review of "Treatment-Resistant Depression in Poland—Epidemiology and Treatment"

_jcm, 2022, doi:10.3390/jcm11030480_

Round 1

Reviewer 1 Report

A very interesting article. However, corrections are needed.

Introduction

There is no general information on epidemiology of treatment resistant depression in the world.

Results

The article lacks the results of statistical tests, e.g. regarding the inequality of groups.

The number of patients should also be given, not just the percentage. This is true for all tables.

In order to improve the quality of the article, the authors could investigate the differences between the different treatment steps (e.g. in terms of treatment).

3.3. - also the lack of adequate statistical analyzes

Author Response

reply to the review attached

Reviewer 2 Report

Very good and well-written paper, and I have only one concern. The authors did not mention (both in the ntroductio and in the Duscussion), that one of the major sources of TRD is sunthreshold bipolarity in the frame of major deoressivev episode (F32 and 33). Several studies show that MDE patients with subthreshold bipolarity show less frequent response to antidepressant monotherapy. For example:

Rybakowski et al Psychopathology, 2007; 40: 153-158

Dudek et a, J Affect Dsord, 2010; 126: 268-271

Woo et al, Int J Psychiat Clin Pract, 2008; 12: 142-146,

Li et al, Brit J Psyíchiat, 2012; 200: 45-51.

and particularly Perugi et al, Eur Neuropsychopharmacology, 2019; 29: 825-834.

If we excludevthese bipolar spectrum patients from the "unipolar" depression group, thec rate of tRD decreases (see: Rihmer et al, Antidepressant response and subthreshold bipolarity in “unipolar” major depressive disorder – Implications for practice and research. Int J Psychopharmacol 2013; 33:  449-452.                                                                                                                                              

The authors aslo should quote the papaer :  Döme P,  et al,  Characteristics of treatment-resistant depression in adults in Hungary: Real-world  evidence from a 7-íear long retrospective data anaqlysis. PLoS ONE 2021, 16(1) e0245510

Author Response

reply to the review attached

Reviewer 3 Report

The paper by Galecki and colleagues aimed to estimate the frequency of treatment resistant depression (TRD) and elucidate the clinical profile of affected patients in outpatient clinics in Poland by asking a representative group of psychiatrists to complete two questionnaires regarding treatment and characteristics of their patients with depression. They found a frequency of TRD of about 25%, which is in line with some findings in the international literature.

The topic of the research is of high relevance to the scientific community in light of the fact that major depressive disorder (MDD) is one of the most frequent and burdensome psychiatric diseases on a global scale and respective information on TRD for Poland is missing. The paper is generally well written, however, it may be further strengthened by considering some points:

  1. Abstract:

1.1 Line 12: …up to 30% of patients fail to achieve remission…

Even though this sentence is not wrong per se, it may be misleading as this third probably refers to later stages in treatment but as the authors surely know, only about 1/3 of MDD patients achieves remission with their first-line antidepressant therapy. This should be clarified also in line 41 of the introduction.

1.2 Line 23 f:The demography, comorbidities, medical history, and pathways of Polish TRD patients were described.

This sentence is not very informative and I don’t understand what the authors mean by “pathways”. Some of the key findings besides the epidemiology should be stated in the abstract as well.

  1. Introduction:

2.1 Line 68: TRD is associated with poorer outcome compared to an MDD that is responsive to treatment.

This sentence can be omitted.

2.2 Line 93: please find another expression for “pathways”.

  1. Material and Methods

3.1 If I understand correctly, only psychiatrist working in outpatient units financed by the National Health Fund were considered. Can the authors please shortly comment on a possible selection bias as theses doctors may see more severely ill patients than psychiatrists working independently in their own practice?

3.2 Did you consider distinguishing between academic and non-academic outpatient clinics for the same reasons mentioned above?

  1. Results

4.1 The authors state that 1,781 patients were described by 76 psychiatrist – in the methods section it says that each psychiatrist provided data for 10-12 patients chosen chronologically which would result in a maximum of 76*12= 912 patients.

Could you please comment.

4.2 The authors have very thoroughly provided information about the prescribed antidepressants in TRD which is interesting. However, the authors should include and describe further therapies (co-medication) that the patients surely received in combination with antidepressants. What about licensed agents for TRD in Europe like Quetiapine XR or Esketamine nasal spray? Some second-generation antipsychotics (aripiprazole, olanzapine, etc…) are licenced in the US for TRD and are widely used off-label in Europe – is this true for Poland. Further interesting agents include benzodiazepines, mood stabilizer (e.g. Lithium), …

4.3. The authors refer to “first therapy and second therapy” etc. I’d suggest to use the terms first-line or second-line antidepressant therapy in the current major depressive episode (MDE) instead.

4.4 Do the authors have information about the dosages of the antidepressants described? It would surely be of interest to the reader.

  1. Discussion

5.1 The authors should consider discussing their findings in light of other European naturalistic TRD samples e.g. those provided by the European Group for the study of resistant depression (GSRD) for instance. As far as I know they used the EMA-definition for TRD as well. Anyway, the comparability may be much better than for the Star*D population mentioned in the introduction.

5.2 Lines 363-364: ..Some studies have suggested that treatment efficacy should be evaluated after 6-8 or even 10-12 weeks…

In fact, newest evidence points towards 2 weeks as optimum time point for the evaluation of treatment response as non-response at this time seems to be predictive of non-response later on as well. The authors may consider to change their arguments in this regard.

5.3 Line 369: “structured interview”

I as I understand the psychiatrist involved filled out questionnaires and did not undergo a structured interview or have I missed something? Please clarify.

  1. Minor Points:

There a couple of minor language mistakes, typos throughout the manuscript that should be corrected accordingly.

Author Response

reply to the review attached

Round 2

Reviewer 3 Report

The authors have thoroughly and adequately addressed all points of criticism in their revised manuscript that I deem very suitable for publication in its present form. I have no further comments.